# Chemo-enzymatic cascades to produce cycloalkenes from bio-based resources

Shuke Wu [1,3]*, Yi Zhou [1], Daniel Gerngross [2], Markus Jeschek [2] & Thomas R. Ward [1]*

Engineered enzyme cascades offer powerful tools to convert renewable resources into value-added products. Man-made catalysts give access to new-to-nature reactivities that may complement the enzyme's repertoire. Their mutual incompatibility, however, challenges their integration into concurrent chemo-enzymatic cascades. Herein we show that compartmentalization of complex enzyme cascades within *E. coli* whole cells enables the simultaneous use of a metathesis catalyst, thus allowing the sustainable one-pot production of cycloalkenes from oleic acid. Cycloheptene is produced from oleic acid via a concurrent enzymatic oxidative decarboxylation and ring-closing metathesis. Cyclohexene and cyclopentene are produced from oleic acid via either a six- or eight-step enzyme cascade involving hydration, oxidation, hydrolysis and decarboxylation, followed by ring-closing metathesis. Integration of an upstream hydrolase enables the usage of olive oil as the substrate for the production of cycloalkenes. This work highlights the potential of integrating organometallic catalysis with whole-cell enzyme cascades of high complexity to enable sustainable chemistry.

[1] Department of Chemistry, University of Basel, Mattenstrasse 24a, BPR 1096, CH-4058 Basel, Switzerland. [2] Department of Biosystems Science and Engineering, ETH Zurich, Mattenstrasse 26, CH-4058 Basel, Switzerland. [3] Present address: Institute of Biochemistry, University of Greifswald, Felix-Hausdorff-Str. 4, D-17489 Greifswald, Germany. *email: shukewu@u.nus.edu; thomas.ward@unibas.ch

The exquisite performance and sophisticated orchestration of metabolic reaction networks have inspired the transition from single-step biocatalysis[1–5] to cascade biocatalysis –interconnected enzymatic reactions in one pot[6,7]. This strategy allows to minimize the isolation and workup of intermediates, thus reducing operation time, waste and, sometimes, enhancing overall selectivity and yield. Furthermore, enzymes may be combined with organometallic catalysts to assemble chemo-enzymatic cascades to perform useful one-pot transformations unattainable with enzymes or small-molecule catalysts alone[8–13]. In the past 6 years, significant progress have been made in combining enzymes with organometallic catalysts[14–21], including photocatalysts[22–24]. For this purpose, different strategies were implemented to accommodate the incompatible reactivities in a single vessel. Despite these achievements, most of the chemo-enzymatic cascades reported to date are limited to one or two enzymatic steps[25,26]. Quite on the contrary, the inherent sophistication of the metabolism in cells, which leverage the exquisite synthetic power of enzyme pathways, suggests the feasibility of much more complex reaction schemes.

Synthetic biology and metabolic engineering hold great promise for sustainable chemical production from renewable resources via engineered microbial cell factories operating in single-vessel processes[27–29]. Combining enzymes from different organisms substantially expands the scope of bioproduction; however, it remains limited to the existing enzymes' repertoire. For example, the recent discovery of decarbonylases and decarboxylases has led to bioproduction of various hydrocarbons[30–32], including various linear alkanes[33,34], linear alkenes[35–37] and aromatics[38,39] (Fig. 1). However, to the best of our knowledge, cycloalkanes and cycloalkenes have not been reported via enzymatic pathways. Cycloalkenes are bulk petrochemicals widely employed in various industrial processes[40], including use as solvents, synthesis of cyclic compounds (e.g. cyclohexanol)[41] and ring-opened chemicals (e.g. adipic acid)[42]. Currently, cycloalkenes are derived from fossil-fuels (e.g. steam cracking of naphtha and partial hydrogenation of benzene)[40]. The unavailability of enzymatic production is likely due to the limited number of enzyme-catalyzed C–C cyclization reactions. Indeed, most of these enzymes[43,44] (e.g. terpene cyclases) are highly substrate-specific thus limiting their widespread implementation. Recent advances in heme-proteins that catalyze an abiotic carbene transfer offer a powerful means to generate cyclopropane or cyclopropene rings (C3)[45–48].

Building on our previous work on ring-closing metathesis (RCM) in a biocompatible environment[49–51], we contemplated the possibility of capitalizing on RCM to construct cycloalkenes (C5-C7) from diolefins produced from renewable sources.

Herein we report on our efforts to combine ring-closing metathesis with an engineered multi-enzyme cascade to produce cyclopentene, cyclohexene, and cycloheptene from olive oil-derived intermediates (Fig. 1). This development, combining a concurrent RCM with up-to nine enzymatic steps hosted by whole cells of *E. coli* in a single reaction vessel, showcases the potential of integrating biocompatible, new-to-nature reactions with synthetic biology at high sophistication.

## Results

**Design of chemo-enzymatic cascades.** We applied a biocatalytic retrosynthetic analysis[52,53] to design chemo-enzymatic cascades to produce cyclopentene (**1a**), cyclohexene (**1b**), and cycloheptene (**1c**) from oleic acid (**6**) (Fig. 2a, b). We hypothesized that cycloalkenes (**1a-1c**) may be accessed via ring-closing metathesis of the corresponding α,ω-dienes (**2a-2c**), which, in turn, can be produced from an oxidative bis-decarboxylation of α,ω-dicarboxylic acids (**4a-4c**) using a decarboxylase[35–37] (Fig. 2a). Exploratory studies using oleic acid (**6**) suggested that decarboxylation of carboxylate **6** affords 1,8-heptadecadiene (**5**), which may undergo ring-closing metathesis to afford cycloheptene **1c** and 1-decene (Fig. 2b). To access the synthetically more useful cyclopentene (**1a**) and cyclohexene (**1b**) from oleic acid **6**, we surmised that one could complement previously reported enzyme cascades[54] –for the production of diacids from oleic acid– with a decarboxylase and an RCM catalyst. The diacids **4a** and **4b** are produced via a six- and four enzyme-cascade, respectively (Fig. 2b). This is achieved via hydration of oleic acid (**6**) by OhyA2, followed by oxidation with MlADH to the corresponding keto-acid. Subjecting the keto-acid to a Baeyer–Villiger mono-oxygenase, using either PpBVMO or PfBVMO, followed by hydrolysis by TLL affords either sebacic acid (**4b**) –with PfBVMO– or a hydroxy-carboxylic acid –with PpBVMO–. The latter may be oxidized by ChnD and ChnE to afford azelaic acid (**4a**)[55] (Fig. 2b, Supplementary Fig. 1 for details). As olive oil is readily hydrolyzed by TLL to afford oleic acid **6**, one may thus be able to produce cycloalkenes **1a-1c**, typically derived from petroleum-based feedstocks, from renewable resources via a concurrent chemo-enzymatic cascade.

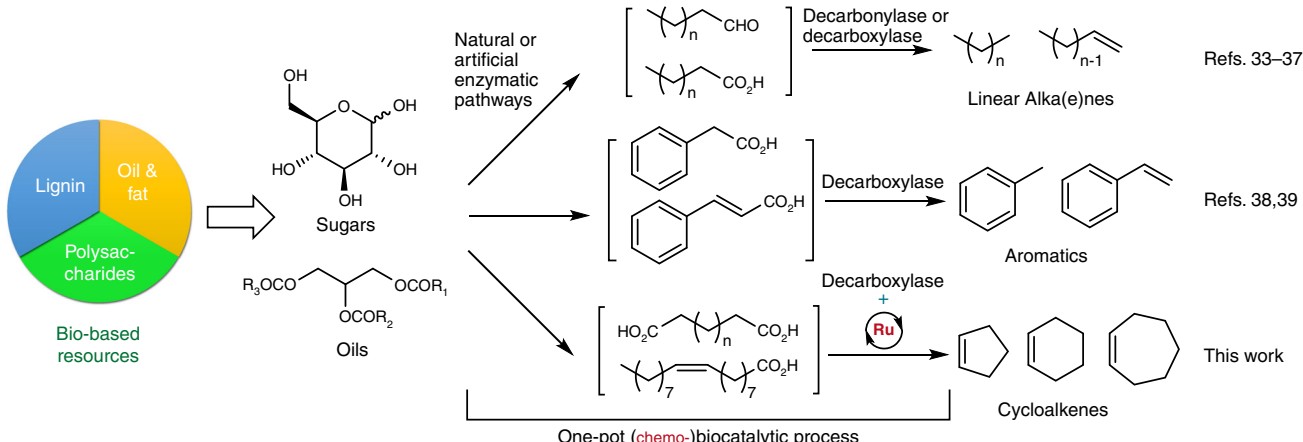

**Fig. 1** Production of hydrocarbons from bio-based resources via (chemo-)enzymatic cascades. Bio-based linear alkanes and alkenes are produced from renewable feedstock via enzyme pathways that include decarbonylases or decarboxylases. Engineering a chemo-enzymatic cascade that comprises a decarboxylase and a ring-closing metathesis catalyst leads to the production of bio-based cycloalkenes

**Fig. 2** Chemo-enzymatic cascades to produce cycloalkenes from dicarboxylic acids, oleic acid, and olive oil. **a** Conversion of diacids (**4a-4c**) to cycloalkenes (**1a-1c**) via a concurrent bis-decarboxylation and metathesis. **b** Chemo-enzymatic cascades for the conversion of olive oil (**7**) or oleic acid (**6**) to cycloheptene (**1c**), cyclopentene (**1a**, via azelaic acid, **4a**) and cyclohexene (**1b**, via sebacic acid, **4b**). **c** Structure of the selected RCM catalyst **Ru3**

**Selection of the decarboxylase**. To identify a suitable enzyme for the bis-decarboxylation of **4a-4c**, we evaluated three oxidative decarboxylases, including: (i) a P450 monooxygenase OleT, combined with the putidaredoxin CamAB[37], (ii) a non-heme mononuclear iron oxidase UndA from *Pseudomonas*[35], and (iii) a membrane-bound desaturase-like enzyme UndB from *Pseudomonas*[36]. These enzymes expressed well in recombinant *E. coli* strains as confirmed by SDS–PAGE analysis of the whole-cell protein extracts (Supplementary Fig. 2). The *E. coli* whole cells were used for the bis-decarboxylation of **4a-4c** and decarboxylation of the intermediate ω-alkenoic acids (**3a-3c**) to afford the corresponding α,ω-dienes (**2a-2c**) in aqueous potassium phosphate buffer (KP). As summarized in Fig. 3a–c, both UndA and UndB displayed good activity towards medium-chain carboxylates **3a-3c**, while OleT was inactive. Remarkably, only UndB catalyzed the bis-decarboxylation of **4a-4c** to afford **2a-2c**. These results are consistent with previous reports: OleT prefers long-chain saturated fatty acids[37,56], while UndA and UndB favor medium-chain fatty acids[35,36]. Under optimized conditions, the bis-decarboxylation of **4a-4c** in the presence of *E. coli* over-expressing UndB was improved to afford increased amounts of **2a** (1600 μM, 32% conversion), **2b** (1900 μM, 38% conversion) and **2c** (800 μM, 16% conversion), respectively (Fig. 3a–c). In view of the hydrophobicity and low aqueous solubility of dienes **2a-2c**, we examined a range of water-miscible and immiscible solvents as well as several surfactants added to the phosphate buffer. The results (Supplementary Fig. 3) suggest that *n*-dodecane (10%), abbreviated KP-Dod hereafter, and the non-ionic surfactant TPGS-750-M[57,58] (1%) are compatible with *E. coli* (UndB), maintaining >50% productivity for the conversion of **4b** to **2b**. Although isooctane is often used in conjunction of purified enzymes[16,17,20], it dramatically reduced the activity of UndB, probably due to the destabilization of the cell membrane. Thus, *n*-dodecane and TPGS-750-M were selected for further studies. Up to ≥80% conversion was obtained starting from 2 mM of **4a** and **4b** or 1 mM of **4c** by *E. coli* (UndB) in KP

buffer with or without either *n*-dodecane or TPGS-750-M (Supplementary Fig. 4).

**Identification of the most suitable metathesis catalyst**. To identify a biocompatible metathesis catalyst, we selected three commercial (Hoveyda)-Grubbs ruthenium(II) catalysts (**Ru1**, **Ru2**, **Ru3**, Supplementary Fig. 5), which have been reported to tolerate air and water[51]. The RCM activity of **Ru1**, **Ru2**, **Ru3** (100 μM, 2 mol%) was evaluated for diolefin **2b** (5 mM) either in: (i) KP buffer, (ii) KP buffer and *n*-dodecane (10%), or (iii) in KP buffer containing the non-ionic surfactant TPGS-750-M (1%) (Fig. 3d). In all cases, the formation of cyclohexene **1b** was detected. The biphasic system with 10% *n*-dodecane clearly out-performed both other systems. To our delight, **Ru3** (Fig. 3d) afforded cyclohexene in >90% conversion for the biphasic system. Both other diolefin substrates **2a** and **2c** afforded cyclopentene **1a** (87% conversion) and cycloheptene **1c** (72% conversion), respectively (Fig. 3e, f). RCM of dienes **2a** and **2b** (5 mM) in the presence of a lower catalyst **Ru3** loading (50 μM, 1%) using the KP-Dod system afforded cycloalkenes **1a** and **1b** in 75% and 88% conversions respectively (corresponding to TTNs of 75 and 88). The commercially available catalyst **Ru3** was thus selected for the implementation of the chemo-enzymatic cascades.

**A cascade to convert diacids to cycloalkenes**. *E. coli* (UndB) whole cells were combined with catalyst **Ru3** for the conversion of sebacic acid **4b** to cyclohexene **1b** in KP buffer via a concurrent chemo-enzymatic cascade. As displayed in Fig. 4a, cyclohexene **1b** (25% conversion) was produced only in the presence of both UndB and catalyst **Ru3**, highlighting the remarkable compatibility of the metathesis catalyst with *E. coli* whole cells. Addition of either *n*-dodecane (10%) or TPGS-750-M (1%) to the reaction mixture led to significantly improved conversions: up to 80% conversion was obtained in the presence of the KP-Dod. The chemo-enzymatic cascade catalysis was also performed in a sequential mode by adding the metathesis catalysts after 24 h

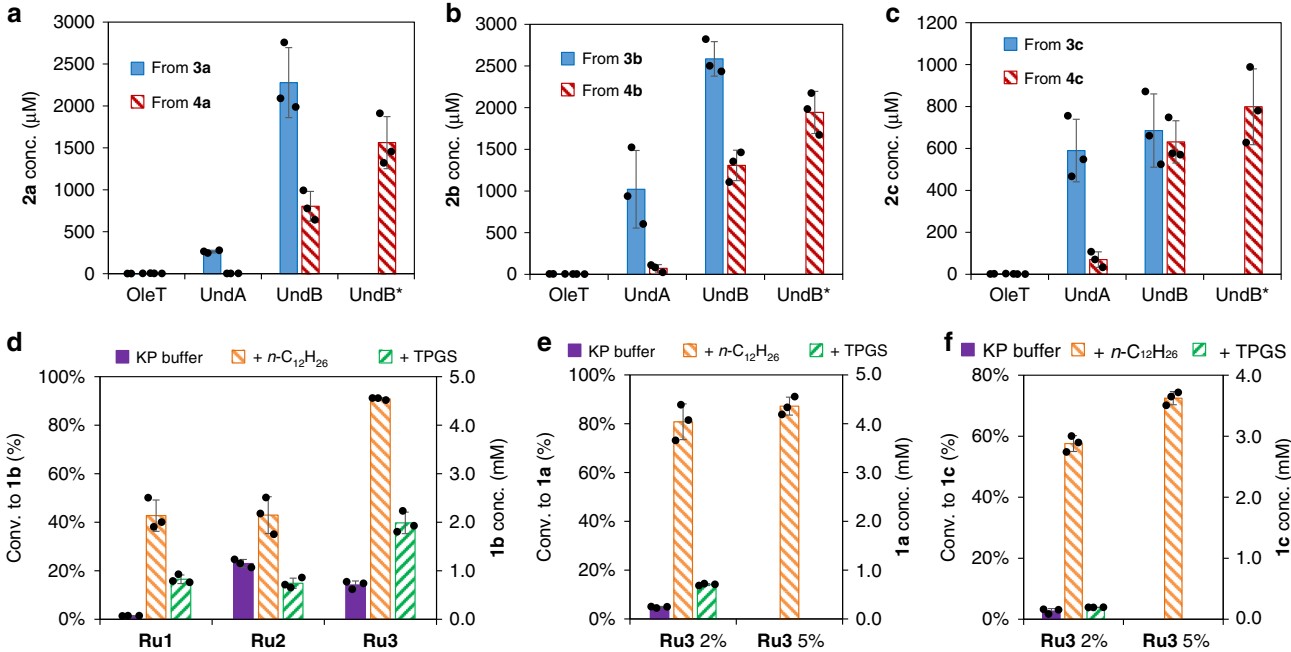

**Fig. 3** Results for the (bis-)decarboxylation by decarboxylases and the RCM by Ru-based catalysts. **a–c** Decarboxylation of **3a-3c** and **4a-4c** (5 mM) to the corresponding dienes **2a-2c** in the presence of *E. coli* cells (10 g l⁻¹) expressing OleT, UndA, or UndB in KP buffer (200 mM, pH 8.0, 2% glucose) at 30 °C for 24 h. * stands for the optimized condition in KP buffer (200 mM, pH 8.0, 1% glucose). **d** RCM of diene **2b** (5 mM) by ruthenium catalyst **Ru1-Ru3** (100 μM) in KP buffer (200 mM, pH 8.0) with or without n-dodecane (10%) or TPGS-750-M (1%) at 30 °C for 24 h. **e, f** RCM of diene **2a** and **2c** (5 mM) by **Ru3** (100–250 μM) in KP buffer (200 mM, pH 8.0). Source data are provided as a Source Data file. Data are mean values of triplicates with error bars indicating standard deviations (*n* = 3). (See Supplementary Fig. 5 for the structure of **Ru1** and **Ru2**)

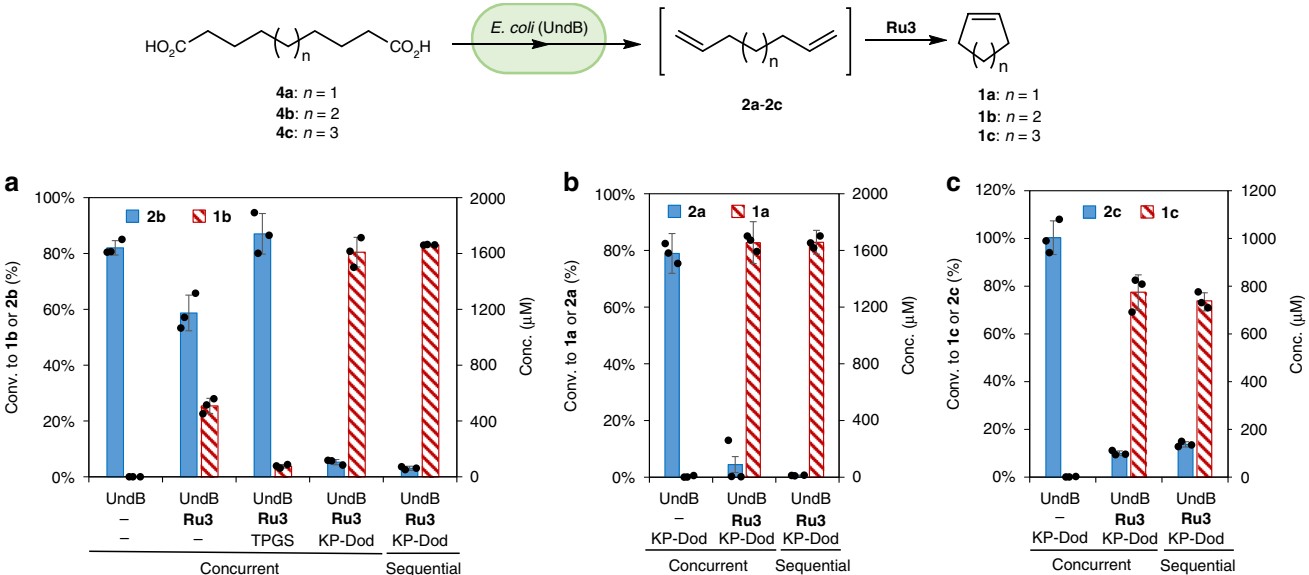

**Fig. 4** Bis-decarboxylation-metathesis cascade for the conversion of diacids (**4a-4c**) to cycloalkenes (**1a-1c**). **a** Concurrent or sequential cascade for the conversion of diacid **4b** (2 mM) to **1b** in the presence of *E. coli* (UndB) (10 g l⁻¹) and **Ru3** (100 μM) in KP buffer (200 mM, pH 8.0, 1% glucose), KP buffer and n-dodecane (10%) or KP buffer and TPGS-750-M (1%) at 30 °C for 24 h. **b** One-pot conversion of diacid **4a** (2 mM) to cyclopentene (**1a**) with *E. coli* (UndB) (10 g l⁻¹) and **Ru3** (100 μM) in KP buffer (200 mM, pH 8.0, 1% glucose) with n-dodecane (10%) at 30 °C for 24 h. **c** One-pot conversion of diacid **4c** (1 mM) to cycloheptene (**1c**) with *E. coli* (UndB) (10 g l⁻¹) and **Ru3** (50 μM) in KP buffer (200 mM, pH 8.0, 1% glucose) with n-dodecane (10%) at 30 °C for 24 h. For the sequential mode, **Ru3** was added at 24 h and reacted for another 24 h (total 48 h). Source data are provided as a Source Data file. Data are mean values of triplicate experiments with error bars indicating standard deviations (*n* = 3)

reaction in the presence of *E. coli* (UndB). The results are similar to those obtained in the concurrent cascade: high conversion (83%) was only achieved with **Ru3** in KP-Dod (Fig. 4a, Supplementary Fig. 6). The sequential cascade however required 48 h instead of 24 h for the concurrent cascade to achieve comparable

conversions. The reaction progress of the concurrent cascade (Supplementary Fig. 7) reveals minimal accumulation of intermediates **2b** and **3b**, thus highlighting that the cascade proceeds in a concurrent manner. Next, the *E. coli* (UndB) and **Ru3** were used in the KP-Dod mixture to convert the diolefins **4a** and **4c** to

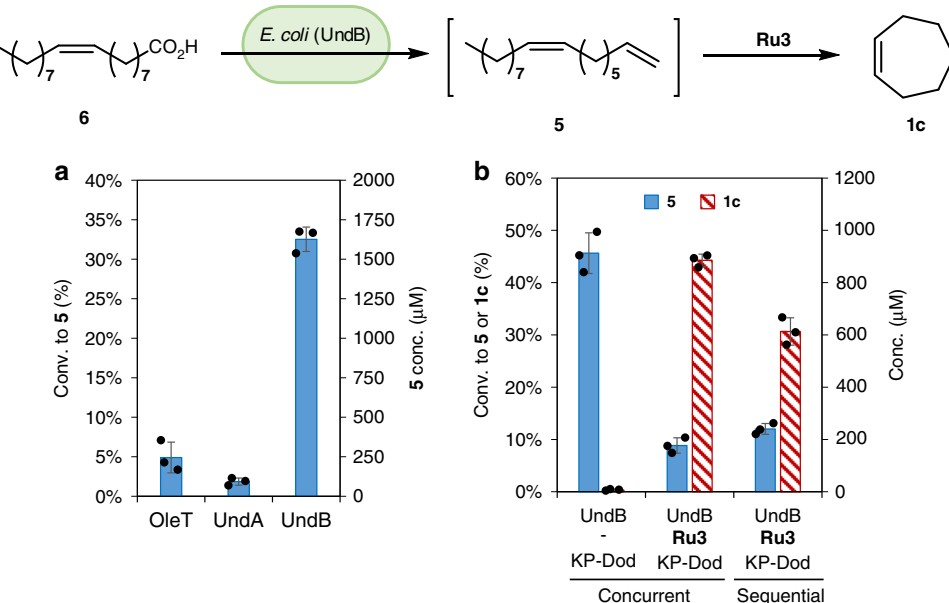

**Fig. 5** Whole-cell decarboxylation-metathesis cascade for the conversion of oleic acid (**6**) to cycloheptene (**1c**). **a** Decarboxylation of oleic acid (**6**) (5 mM) to diolefin **5** by *E. coli* cells (10 g l⁻¹) expressing OleT, UndA, or UndB in KP buffer (200 mM, pH 8.0, 1% glucose) at 30 °C for 24 h. **b** Concurrent conversion of oleic acid (**6**) (2 mM) to cyclopentene **1c** with *E. coli* (UndB) (10 g l⁻¹) and **Ru3** (100 µM) in KP buffer (200 mM, pH 8.0, 1% glucose) in the presence of *n*-dodecane (10%) at 30 °C for 24 h. For the sequential cascade, *n*-dodecane and **Ru3** were added at 12 h and reacted for another 12 h (total 24 h). Source data are provided as a Source Data file. Data are mean values of triplicate experiments with error bars indicating standard deviations (*n* = 3)

cyclopentene **1a** and cycloheptene **1c** in high conversion (83% and 77% conversion, respectively, Fig. 4b, c).

**A concurrent cascade to convert oleic acid to cycloheptene.** Upon enzymatic decarboxylation, oleic acid (**6**) is converted to 1,8-heptadecadiene (**5**), a bio-based precursor of cycloheptene (**1c**, Fig. 2b). Accordingly, a two-step decarboxylation-metathesis cascade was engineered to convert oleic acid (**6**) to cycloheptene (**1c**). The *E. coli* strains over-expressing OleT, UndA, and UndB were examined for decarboxylation of oleic acid (**6**) (5 mM) in KP buffer, to afford 1,8-heptadecadiene (**5**). While OleT is known to display very modest activity toward Δ⁹ unsaturated fatty acids[59], *E. coli* (UndB) performed the best, giving 33% conversion to diene **5** (Fig. 5a). Upon combining *E. coli* (UndB) with **Ru3**, oleic acid (**6**) (2 mM) was converted to cycloheptene (**1c**) in the KP-Dod system in 44% conversion (Fig. 5b). The chemo-enzymatic cascade also performed well in a sequential mode: adding the catalyst **Ru3** after 12 h allowed to maintain the same overall reaction time at the cost of a lower conversion however (24 h, 33% conversion, Fig. 5b). In a similar fashion, the two-step chemo-enzymatic cascade could be used to produce cyclopentene (**1a**) and cyclohexene (**1b**), but the corresponding fatty acids with a C=C bond at position 7 and 8 are not readily available. To access these cycloalkenes, we thus engineered the corresponding enzyme cascades in *E. coli* to produce the diolefins from oleic acid (**6**).

**Engineering *E. coli* cells to convert oleic acid to diacids.** To produce cyclopentene (**1a**) and cyclohexene (**1b**) from oleic acid (**6**), it is necessary to engineer enzyme cascades to produce the corresponding sebacic acid (**4b**) and azelaic acid (**4a**) from oleic acid (**6**). Two previous enzyme cascades[54] have reported the conversion of oleic acid (**6**) to diacids **4a** and **4b** via hydration, oxidation, Baeyer–Villiger oxidation (with different regioselectivity) and hydrolysis by TLL (for **4b**), as well as further double oxidation (for **4a**)[55] (Fig. 2b, Supplementary Fig. 1 for details). The reported cascades were accomplished by combining multiple

*E. coli* strains as well as purified enzymes. We set out to implement this complex cascade in a single *E. coli* strain.

According to previous reports[54,55] and our experience, the bottleneck in the above cascades is likely the Baeyer–Villiger oxidation by PfBVMO (for **4b**) and PpBVMO (for **4a**). Some N-terminal tags on BVMOs have been shown to improve the stability of the enzymes[60]. Thus, fusion of BVMOs with different N-terminal tags was tested: the 6x Glu-tagged PfBVMO and 6x His-tagged PpBVMO outperformed other constructs, including BVMOs with no tag (Supplementary Figs. 8 and 9). A native fatty-acid transporter, FadL, was overexpressed and found to boost the hydration of oleic acid (**6**) in *E. coli* (Supplementary Fig. 10). The other enzymes (OhyA2, MlADH, TLL) expressed well in individual *E. coli* strains (Supplementary Fig. 11).

In order to combine all required activities for the conversion of oleic acid (**6**) to the dicarboxylate **4a** or **4b** in a single *E. coli* strain, we sought to optimize the absolute and relative expression levels of the corresponding enzymes relying on engineering of the respective ribosome binding sites (RBSs). Therefore, we performed combinatorial pathway optimization by applying the recently developed algorithm RedLibs[61] to design tailor-made degenerate RBS sequences. Briefly, degenerate RBSs were designed to achieve four different expression levels for each of the four involved genes with the goal to uniformly span the accessible range (Supplementary Fig. 20). For the production of sebacic acid (**4b**), primers with degenerate RBS (4×) were used to amplify PfBVMO, MlADH, OhyA2, and TLL (with the plasmid backbone). The resulting combinatorial library (combinatorial size: 4 × 4 × 4 × 4 = 256 variants) was assembled *in vitro* and used to transform *E. coli* (Fig. 6a). In total 576 of the resulting colonies were randomly picked (~89.5% statistical coverage), cultured in six 96-deep well plates and assayed for the conversion of oleic acid (**6**). The concentration of sebacic acid (**4b**) was quantified by UPLC analysis (Fig. 6b). The best four strains for each plate were cultured in flasks for further validation and quantification of the produced sebacic acid (**4b**) (Fig. 6c). The most promising strain, coined *E. coli* (C10) hereafter (P1-A4 in Fig. 6c), was selected for converting oleic acid (**6**) to sebacic acid (**4b**) in further

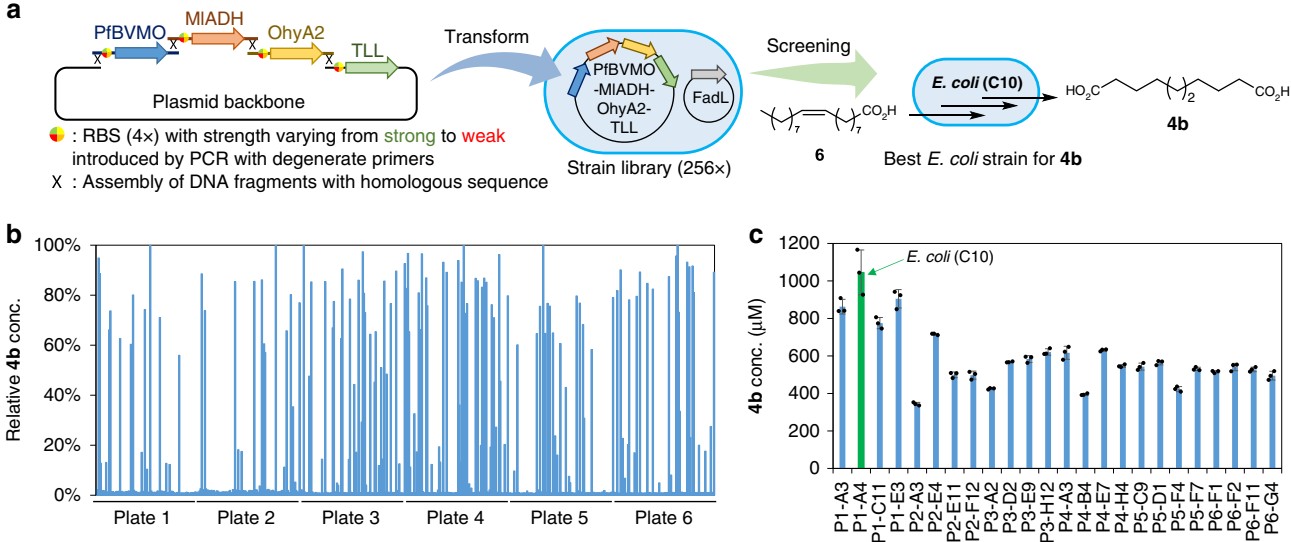

**Fig. 6** Engineering and screening a strain library for the production of sebacic acid (**4b**) from oleic acid (**6**). **a** Construction of a plasmid library for the co-expression of PfBVMO, MlADH, OhyA2, and TLL with different expression levels, and screening of the resulting *E. coli* library to identify the most effective strain for the production of sebacic acid (**4b**). All *E. coli* strains included an additional plasmid harboring the fatty acid transporter FadL. **b** Initial screening of 576 strains in six 96-well plates for production of sebacic acid (**4b**). **c** Further validation and comparison of 24 strains (best four from each plate) for the production of sebacic acid (**4b**). The green column indicates the best strain *E. coli* (C10). The reactions were performed using oleic acid (**6**) (5 mM) and *E. coil* (10 g l$^{-1}$) in KP buffer (200 mM, pH 8.0, 1% glucose) at 30 °C for 24 h. Source data are provided as a Source Data file. Data in **c** are mean values of triplicate experiments with error bars indicating standard deviations (n = 3)

experiments. For the more complex production of azelaic acid (**4a**), a two-stage optimization procedure using RedLibs was applied to construct and identify the best strain, *E. coli* (C9). To this end, we first optimized PpBVMO, MlADH, OhyA2, and TLL for production of 9-hydroxynonanoic acid in a first combinatorial RBS optimization similar as before for the production of sebacic acid (Supplementary Fig. 12). Afterwards, the best-performing clone from this first stage was applied to a second round of combinatorial screening to optimize ChnD, ChnE, and FadL for production of azelaic acid (**4a**) (Supplementary Fig. 13).

**Cascades for oleic acid to cyclopentene and cyclohexene.** Cyclopentene (**1a**) may be produced from oleic acid (**6**) via a chemo-enzymatic cascade by combining *E. coli* (C9), *E. coli* (UndB) and **Ru3** either in a concurrent- or a sequential cascade. Since *E. coli* (UndB) also catalyzes the decarboxylation of oleic acid (**6**) to form the diolefin 1,8-heptadecadiene (**5**), *E. coli* (UndB) and **Ru3** can only be added *after* the conversion of oleic acid to the corresponding diacids, sebacic acid (**4b**) and azelaic acid (**4a**), respectively. As highlighted in Fig. 7a, adding all three catalytic components simultaneously (*i.e. E. coli* (C9), *E. coli* (UndB) and **Ru3**), cyclopentene (**1a**) was produced in only 14% conversion from oleic acid **6** (2 mM). In contrast, the delayed addition of *E. coli* (UndB) and **Ru3** led to cyclopentene (**1a**) in 65% conversion under the same reaction conditions. We surmise that the cross-reactivity of *E. coli* (UndB) is the major ground for the erosion of the conversion in the concurrent cascade. For the production of cyclohexene (**1b**) from oleic acid (**6**), the *E. coli* (C10), and *E. coli* (UndB) strains as well as the RCM catalyst **Ru3** were combined in one pot either simultaneously or sequentially, (Fig. 7b). Again here, the delayed addition of *E. coli* (UndB) and **Ru3** led to significantly higher conversions of cyclohexene (**1b**) (compare 6 % for the simultaneous *vs.* 22 % conversion for the sequential cascade starting with 2 mM oleic acid (**6**)). The modest conversion of cyclohexene (**1b**) is tentatively assigned to the low productivity of PfBVMO (Supplementary Fig. 14). The modest productivity of the former may be improved by engineering a

more active BVMO. In summary, by combining two engineered *E. coli* strains and a ruthenium complex **Ru3**, cyclopentene (**1a**) and cyclohexene (**1b**) were produced from oleic acid (**6**) in one pot in titers 1.30 mM and 0.44 mM after 24 h, respectively.

**Cascades to convert olive oil to cycloalkenes.** Currently, oleic acid is usually manufactured from plant oils (e.g. canola oil, olive oil) on a very large scale[62]. To directly utilize these attractive bio-based feedstocks (or even waste oil), we further extend the chemo-enzymatic cascades from a readily available plant oil–olive oil (**7**), as an example. The enzymatic hydrolysis of olive oil (**7**) produces oleic acid (**6**) under mild conditions and is compatible with the implementation of enzyme-cascades. We thus engineered *E. coli* (TLL) cells expressing the lipase from *Thermomyces lanuginosus* for the hydrolysis of olive oil (**7**) to oleic acid (**6**). The *E. coli* (TLL) cells were lyophilized and stored. Using the lyophilized cells (1 g l$^{-1}$), oleic acid (**6**) was produced with titers ranging from 1.1-2.5 mM from olive oil **7** (1-2 g l$^{-1}$) in 1–3 h (Supplementary Fig. 15). To produce cycloheptene (**1c**) from olive oil **7**, lyophilized *E. coli* (TLL), *E. coli* (UndB) and **Ru3** were combined simultaneously or sequentially in one pot. As presented in Fig. 8a, 610–670 µM of cycloheptene (**1c**) was produced from olive oil (**7**, 1 g l$^{-1}$) via this chemo-enzymatic cascade in a concurrent mode as well as two complementary sequential modes. A third sequential cascade afforded cycloheptene (**1c**) in lower concentration (330 µM). We hypothesize that the lower conversion may be due to the limited RCM reaction time applied in the third sequential cascade. For the production of cyclopentene (**1a**) from olive oil (**7**), lyophilized *E. coli* (TLL), *E. coli* (C9), *E. coli* (UndB) and **Ru3** were combined sequentially in one pot: to our delight, 760 µM of cyclopentene (**1a**) was produced from olive oil (**7**, 1 g l$^{-1}$) via this extended chemo-enzymatic cascade that includes nine enzymatic reactions preceding the final ring-closing metathesis step (Fig. 8b). Similarly, combining lyophilized *E. coli* (TLL), *E. coli* (C10), *E. coli* (UndB) and **Ru3** sequentially, enabled the production of cyclohexene (**1b**) in 710 µM from olive oil (**7**, 1 g l$^{-1}$) via a chemo-enzymatic cascade that includes seven

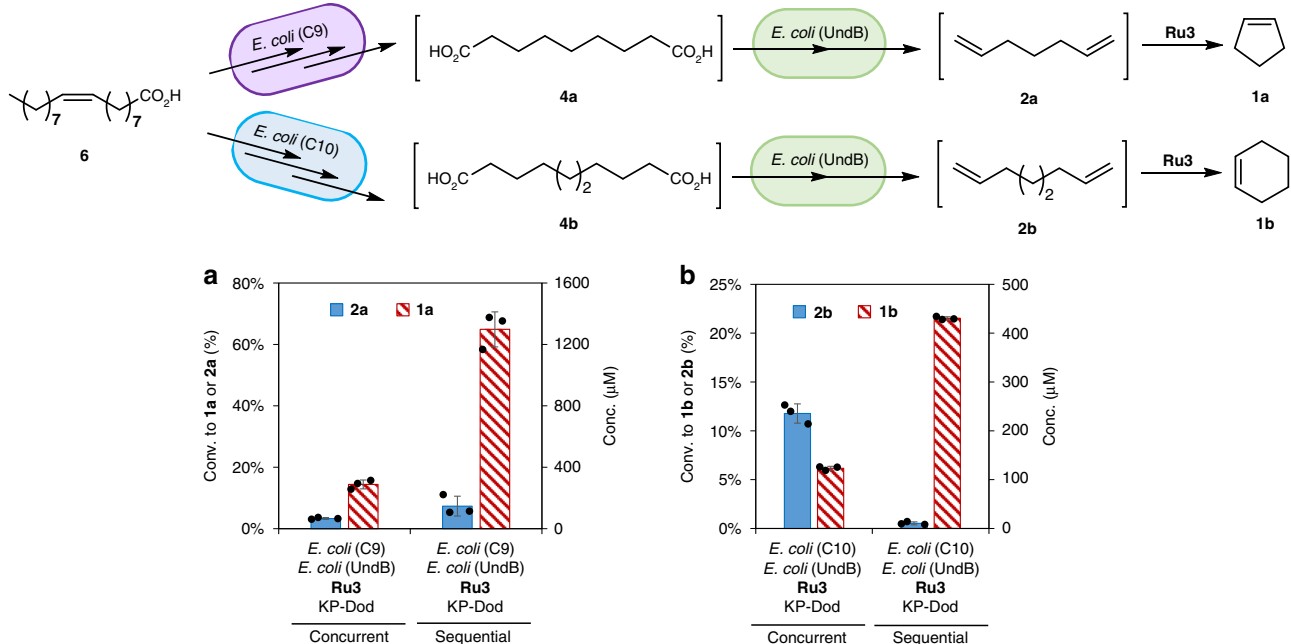

**Fig. 7** Chemo-enzymatic cascades for the conversion of oleic acid to cyclopentene (**1a**) and cyclohexene (**1b**). **a** One-pot conversion of oleic acid (**6**) (2 mM) to cyclopentene (**1a**) using *E. coli* (C9) (10 g l$^{-1}$), *E. coli* (UndB) (10 g l$^{-1}$) and **Ru3** (100 μM) in KP buffer (200 mM, pH 8.0, 1% glucose) with *n*-dodecane (10%) at 30 °C for 24 h. **b** One-pot conversion of oleic acid (**6**) (2 mM) to cyclohexene (**1b**) using *E. coli* (C10) (10 g l$^{-1}$), *E. coli* (UndB) (10 g l$^{-1}$) and **Ru3** (100 μM) in KP buffer (200 mM, pH 8.0, 1% glucose) with *n*-dodecane (10%) at 30 °C for 24 h. For the sequential cascade, *E. coli* (UndB), **Ru3** and *n*-dodecane were added after 12 h and the reaction was carried on for another 12 h (total 24 h). Source data are provided as a Source Data file. Data are mean values of triplicate experiments with error bars indicating standard deviations (n = 3)

enzymatic steps preceding the final RCM reaction (Fig. 8c). Assuming oleic acid accounts for 70% of the fatty acids in olive oil, the production of cycloalkenes (**1a-1c**) in 670-760 μM corresponds to around 29–33% conversion from olive oil (**7**, 1 g l$^{-1}$). Although the overall conversion to cycloalkenes is rather modest, the chemo-enzymatic cascades showcase the feasibility to produce cycloalkenes from a renewable feedstock. Currently, the performance of the cascades is limited by the modest decarboxylase activity of UndB, which we surmise may be overcome by directed evolution[63–67]. Considering that oleic acid and diacids are often encountered by microbes, discovering and evolving yet unknown decarboxylases specific for oleic acid or diacids may allow to overcome the current bottleneck of the cascades. In analogy to the discovery of an efficient decarboxylase[38] which led to the efficient production of styrene via a two-step deamination-decarboxylation cascade[68], the discovery and/or evolution of a more efficient decarboxylase may allow to significantly enhance the yields of cycloheptene **1c** via the two-step decarboxylation-metathesis cascade. Similarly, the extended cascades towards cyclopentene **1a** and cyclohexene **1b**, may be further improved by relying on recently evolved[69] or discovered[70] BVMOs. Nevertheless, the extended one-pot chemo-enzymatic cascades that include (up to) nine enzymatic steps demonstrates the possibility to integrate a transition metal-catalyzed reaction with synthetic biology at high complexity.

## Discussion

In summary, we have integrated a ruthenium-catalyzed ring-closing metathesis reaction as the last step of complex enzyme cascades for the one-pot production of cyclopentene (**1a**), cyclohexene (**1b**), and cycloheptene (**1c**) from olive oil (**7**) and its derivatives. The development of an aqueous-*n*-dodecane two-phase system enabled the combination of a ruthenium-based catalyst **Ru3** with a delicate membrane-bound decarboxylase as part of extended enzyme cascades in *E. coli* whole cells, including

up to nine enzymatic steps. The (chemo)-bioproduction of cycloalkenes from bio-based resources significantly expands the product realm accessible thanks to synthetic biology. The efficiency of the chemo-enzymatic cascades may be further engineered via the identification and/or the directed evolution of (more active) enzymes. The complementarity of homogeneous catalysts and enzymes allows the introduction of new-to-nature reactions with whole-cell enzyme cascades, thus synergizing the synthetic power of organometallic chemistry and synthetic biology.

## Methods

**Materials**. **Ru1**, **Ru2**, **Ru3**, and *n*-dodecane were purchased from Sigma-Aldrich and used without further purification. Except for OleT, the genes of all the other enzymes were synthesized from Gene Universal. The recombinant *E. coli* strains were engineered using standard molecular cloning techniques by restriction enzymes or Gibson assembly. All the other chemicals were purchased from commercial suppliers and used without further purification. See Supplementary Table 1 for all the primers used in this study.

**Preparation of engineered *E. coli* whole cells**. LB medium (1 ml) with appropriate antibiotics was inoculated with recombinant *E. coli* cells and cultured for 8–10 h at 37 °C 250 rpm. Then, the cells were transferred to modified M9 medium (50 ml, with 2% glucose and 0.6% yeast extract) in a baffled flask to grow at 37 °C until OD$_{600}$ of the culture reached 0.6–0.8. At this time, IPTG was added to a final concentration of 0.5 mM. The culture was carried on at 22 °C for 12–14 h. The *E. coli* cells were harvested by centrifugation and immediately used in the enzyme cascades. For *E. coli* (TLL), the cells were harvested, washed with water, lyophilized, stored at 4 °C, and added as powder when needed.

**Typical procedure for performing chemo-enzymatic cascades**. Substrate stock solutions were prepared by dissolving in EtOH: **4a**-**4c** (250 mM), **6** (250 mM), and **7** (100 g l$^{-1}$, emulsion). Stock solutions of cells were prepared by resuspending freshly harvested *E. coli* (C9), *E. coli* (C10) and/or *E. coli* (UndB) cells in KP buffer (200 mM, pH 8.0). The density of cells (OD$_{600}$) was determined with a UV-spectrometer. Stock solutions of **Ru3** (5 mM) were freshly prepared by dissolving **Ru3** in DSMO: EtOH 1: 1. To an air-tight reaction tube (25 ml) with screw-cap, stock solutions of cells, KP buffer (200 mM, pH 8.0) and stock solution of glucose (50%) were added to form a system (0.5 ml) with required density/concentrations

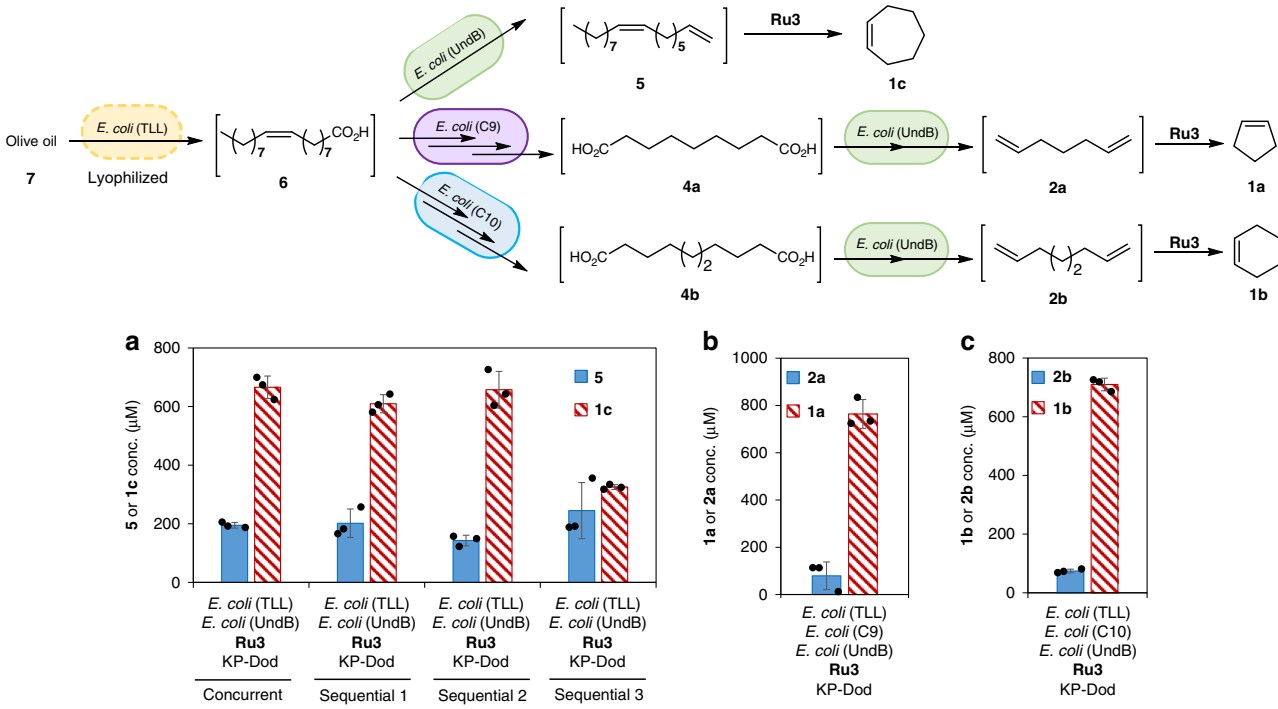

**Fig. 8** Chemo-enzymatic cascades for the conversion of olive oil (**7**) to cycloalkenes (**1a-1c**). **a** One-pot conversion of olive oil (**7**) (1 g l$^{-1}$) to cyclopentene (**1a**) with *E. coli* (TLL) (1 g l$^{-1}$), *E. coli* (UndB) (10 g l$^{-1}$) and **Ru3** (100 µM) in KP buffer (200 mM, pH 8.0, 1% glucose) in the presence of *n*-dodecane (10%) at 30 °C for 24 h. Sequential 1: **Ru3** was added after 2 h (total reaction time: 24 h). Sequential 2: *E. coli* (UndB) and **Ru3** were added after 2 h (total 24 h). Sequential 3: *E. coli* (UndB) was added after 2 h, and **Ru3** was added after 12 h (total 24 h). **b** One-pot conversion of **7** (1 g l$^{-1}$) to cyclopentene (**1a**) with *E. coli* (TLL) (1 g l$^{-1}$), *E. coli* (C9) (10 g l$^{-1}$, added after 1 h), *E. coli* (UndB) (10 g l$^{-1}$, added after 12 h) and **Ru3** (100 µM, added after 12 h) in KP buffer (200 mM, pH 8.0, 1% glucose) with *n*-dodecane (10%, added after 12 h) at 30 °C for 24 h. **c** One-pot conversion of olive oil (**7**) (1 g l$^{-1}$) to cyclohexene (**1b**) with *E. coli* (TLL) (1 g l$^{-1}$), *E. coli* (C10) (10 g l$^{-1}$, added after 1 h), *E. coli* (UndB) (10 g l$^{-1}$, added after 12 h) and **Ru3** (100 µM, added after 12 h) in KP buffer (200 mM, pH 8.0, 1% glucose) with *n*-dodecane (10%, added after 12 h) at 30 °C for 24 h. Source data are provided as a Source Data file. Data are mean values of triplicate experiments with error bars indicating standard deviations (*n* = 3)

(cells: 10 g l$^{-1}$, glucose: 1%). Then, *n*-dodecane (50 µl) and stock solutions of **Ru3** (5–10 µl) were added to the reaction tube. The stock solutions of the substrates (2–10 µl) were added last. The reaction tubes were sealed and incubated at 250 rpm, 30 °C for 24 h. For the cascades in the sequential mode, the appropriate amounts of cells, *n*-dodecane (50 µl) and **Ru3** were added at the specified time. Upon completion, the reaction tubes were incubated on ice for 15–20 min before opening these (to minimize the loss of volatile alkenes) and adding of EtOAc (1 ml, containing 1 mM of acetophenone as internal standard). The reaction products were extracted, dried over Na$_2$SO$_4$, and analyzed by GC–MS. See Supplementary Methods for details. See Supplementary Figs. 16 and 21 for the calibration curves. See Supplementary Figs. 17–19 for the representative GC-MS chromatograms.

**Reporting summary**. Further information on research design is available in the Nature Research Reporting Summary linked to this article.

## Data availability

The DNA sequences of the synthetic genes are available in the SI file and they have been deposited in GenBank with the accession codes (UndA: MN125553; UndB: MN125554; PfBVMO: MN125555; PpBVMO: MN125556; MlADH: MN125557; OhyA2: MN125558; TLL: MN125559; ChnD: MN125560; ChnE: MN125561). The source data used to generate Figs. 3–8 and Supplementary Figs. 2–4, 6–16 and 20–21 are provided as a Source Data file. The other data that support the findings of this study are available from the corresponding authors upon request.

## Code availability

The source code for RedLibs algorithm is available at GitHub: https://github.com/dgerngross/RedLibs.

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

## Acknowledgements
We thank K. Faber, A. Dennig, and A. Schallmey for sharing the plasmids pET28a-OleT and pACYC-CamAB. T.R.W. thanks generous support from Swiss National Science Foundation (Grant SNF 200020_162348), the ERC advanced grant (the DrEAM, grant agreement No 694424) and the NCCR Molecular Systems Engineering. S.W. thanks the Federal Commission for Scholarships for Foreign Students for an ESKAS Scholarship.

## Author contributions
T.R.W. and S.W. conceived and designed the project. S.W. and Y.Z. performed the experiments and analyzed the results. M.J. and D.G. designed RBS libraries in silico. T.R.W. supervised the whole project. S.W. and T.R.W. wrote the manuscript with inputs from all authors.

## Competing interests
The authors declare no competing interests.
