## [Peer Review File · Nature Communications]

Reviewers' comments:

Reviewer #1 (Remarks to the Author):

In the revised manuscript, Ward et al. have addressed all minor points raised by me in a sufficient manner. Yet, I still have hesitations about the ground-breaking character of the manuscript that would justify publication in a Nature journal.

The main limitation of the method is the rather low activity of oxidative fatty acid decarboxylases in cellular systems. Here, several groups attempted to convert different fatty acids using bacterial and fungal hosts, but the yields were limited to a few mg/L. This is a big difference to terpenoids, where often a few g/L can be reached. Overcoming the productivity limitation of cellular fatty acid decarboxylation would be a breakthrough, but the present manuscript does not offer any new approach to overcome this critical challenge. The authors state that the low performance of whole-cell catalysts in the oxidative decarboxylation of fatty acids can be simply overcome by directed evolution of the decarboxylase. I do not find this convincing as it does not take into account crucial metabolic aspects such as the low intracellular availability of the toxic free fatty acids in the cytosol.

Despite the high scientific quality, the present manuscript represents a combination of already established reactions and cascades rather than a breakthrough or a new concept. For the claimed application for the synthesis of bio-based chemicals, yield and productivity are several order of magnitudes too low, and the paper does not offer a convincing perspective that this can be substantially improved in the future. As the combination of different enzymatic reactions in the low mM-range with a chemical reaction is of high technical interest, the manuscript is more suitable for a chemical journal such as ACS catalysis.

Reviewer #2 (Remarks to the Author):

The authors have addressed some of the issues that were raised by the three reviewers. Doing this the authors have slightly improved the manuscript. However, the major concern of both, reviewer 1 and 2, that the work only represents some optimization/slight improvements of existing technologies and reactions that have been shown extensively before has not been addressed. The yield and titers are very low. Apparently, as the authors also write in their reply, yield and titer even decrease further with increasing substrate concentration. Unfortunately no clear vision our route is given in the manuscript that would alleviate this. For example, the authors now write "Directed evolution efforts⁶³⁻⁶⁷ should undoubtedly allow improving the catalytic efficiency and turnover numbers of the individual enzymes used herein."

So far directed evolution approaches gave only very modest improvements of OleT. "should undoubtedly" therefore does not appear to be valid.

The point regarding the very low TTN of the Ruthenium catalyst (below 10) was not addressed.

The question of mass balance was not addressed sufficiently. Indeed there are unavoidable losses; however, these become significant simply because the total conversion is extremely low. If titers and yields were much higher these losses weren't significant anymore.

Taken together:

I still believe the manuscript to be impressive by the sheer amount of good work performed and that it deserves publication in a higher impact journal. I doubt it to be sufficient for Nature Communications due to the missing overarching novelty of the concept and the meager yields and titers.

Reviewer #3 (Remarks to the Author):

In this paper, Wu, Ward and co-workers report the development of three one-pot chemoenzymatic cascades to produce cyclopentene, cyclohexene, and cycloheptene. These routes take advantage of a renewable starting material, whole cell biocatalysis, and a biocompatible organometallic catalyst. The goal of this work is to sustainably produce cycloalkenes that are traditionally sourced from petroleum, and to further demonstrate that an organometallic catalyst can be interfaced with a multi-enzyme cascade. The authors demonstrate that most individual whole-cell biocatalytic steps can be optimized with various organic or detergent additives, or as protein fusion constructs. They also find that concurrent or sequential addition of multiple whole-cell biocatalysts and/or cross-metathesis catalysts can improve conversion. Finally, the authors engineer and screen plasmids encoding for several enzymes that contain various combinations of ribosome binding sites in order to optimize the steps from oleic acid to the required dicarboxylic acids of different chain length in a single strain of *E. coli*.

Overall, this study is well suited for Nature Communications. The results reported will be appealing to a wide scientific audience interested in green chemistry, biocatalysis, metabolic engineering, biocompatible chemistry, and/or catalysis more broadly. In particular, this work is notable because of the high number of steps incorporated into the chemoenzymatic cascades, as well as the creative integration of alkene-forming enzymatic decarboxylation reactions with ring-closing metathesis. The results are succinctly presented and discussed in a scholarly manner, and the documentation of experiments is complete. Furthermore, the authors have satisfactorily addressed reviewer concerns from the previous submission. Thus, I recommend accepting this paper.

Reviewer #4 (Remarks to the Author):

The authors have fully addressed all of my concerns and I think this manuscript can now be accepted for publication.

Point-by-point response to the referee's comments (NCOMMS-19-23255-T)

Reviewer #1 (Remarks to the Author):

1.1) The main limitation of the method is the rather low activity of oxidative fatty acid decarboxylases in cellular systems. Here, several groups attempted to convert different fatty acids using bacterial and fungal hosts, but the yields were limited to a few mg/L. This is a big difference to terpenoids, where often a few g/L can be reached. Overcoming the productivity limitation of cellular fatty acid decarboxylation would be a breakthrough, but the present manuscript does not offer any new approach to overcome this critical challenge.

Reply: We agree with the reviewer that the rather low activity of fatty acid decarboxylases is a general limitation in the field. However, the novelty of our work is the integration of an organometallic catalyst with whole-cell enzyme cascades of unprecedented complexity. This allowed us to access a group of important chemicals (cycloalkenes, inaccessible with enzymes alone) from renewable resources. We discuss potential approaches to further improve the decarboxylases and the chemo-enzyme cascades (see reply to point 1.2 below).

1.2) The authors state that the low performance of whole-cell catalysts in the oxidative decarboxylation of fatty acids can be simply overcome by directed evolution of the decarboxylase. I do not find this convincing as it does not take into account crucial metabolic aspects such as the low intracellular availability of the toxic free fatty acids in the cytosol.

Reply: We thank the reviewer for this comment. We think the modest activity of decarboxylation of fatty acids may be overcome by directed evolution of existing decarboxylases or the discovery of new natural decarboxylases specific for oleic acid and diacids.

The following two sentences were added to page 16:

Currently, the performance of the cascades is limited by the modest decarboxylase activity of UndB, which we surmise may be overcome by directed evolution⁶³⁻⁶⁷. Considering that oleic acid and diacids are often encountered by microbes, discovering and evolving yet unknown decarboxylases specific for oleic acid or diacids may allow to overcome the current bottleneck of the cascades.

The low intracellular availability of free fatty acids has been (partially) addressed by introducing a fatty acid transporter (FadL, see page 11 and Fig. 6a). FadL significantly boosted the hydration of oleic acid in *E. coli* (up to 7 times for the initial 30-min reaction, see Supplementary Fig 10) by increasing the transport of oleic acid into cells. We feel that the toxicity or the metabolic aspects of fatty acids is less relevant to our study, as the cascade reactions were performed with resting/non-growing *E. coli* cells (oleic acid is supplied externally, the cells are less influenced by the toxicity of oleic acid). This contrasts with a fermentation process using growing cells (where the oleic acid is produced by the cellular metabolism).

1.3) Despite the high scientific quality, the present manuscript represents a combination of already established reactions and cascades rather than a breakthrough or a new concept. For the claimed application for the synthesis of bio-based chemicals, yield and productivity are several orders of magnitude too low, and the paper does not offer a convincing perspective that this can be substantially improved in the future. As the combination of different enzymatic

reactions in the low mM-range with a chemical reaction is of high technical interest, the manuscript is more suitable for a chemical journal such as ACS catalysis.

Reply: We agree that the yield and productivity of the chemo-enzymatic cascades are currently modest. However, the efficiency may be significantly improved once an efficient decarboxylase is engineered or discovered.

The possibility has been demonstrated for the bioproduction of styrene: styrene was initially produced via deamination-decarboxylation in 260 mg/L (2.5 mM) in a first report (*Metab. Eng.* **2011**, *13*, 544). Subsequently, an important component of the decarboxylase was discovered (Ref 38, *Nature* **2015**, *522*, 497). Relying on this component and a more efficient decarboxylase, styrene could be produced in 14.5 g/L (139 mM) by the two-step deamination-decarboxylation cascade (Ref 68, *Angew. Chem. Int. Ed.* **2016**, *55*, 11647).

In addition, the extended cascades may be also limited by the efficiency of BVMOs. Again here, this bottleneck may be lifted either by directed evolution and/or the discovery of more efficient BVMOs as demonstrated recently (Ref 69 & 70, *J. Am. Chem. Soc.* **2018**, *140*, 10464; *ChemBioChem*, **2018**, *19*, 2049).

The following text was added on page 16-17, and Ref 68-70 were included:

In analogy to the discovery of an efficient decarboxylase³⁸ which led to the efficient production of styrene via a two-step deamination-decarboxylation cascade,⁶⁸ the discovery and/or evolution of a more efficient decarboxylase may allow to significantly enhance the yields of cycloheptene **1c** via the two-step decarboxylation-metathesis cascade. Similarly, the extended cascades towards cyclopentene **1a** and cyclohexene **1b**, may be further improved by relying on recently evolved⁶⁹ or discovered⁷⁰ BVMOs.

We believe that this study is well suited for *Nature Communications* as it may be of interest to a wide audience spanning biocatalysis, green/sustainable chemistry, biocompatible chemistry, synthetic biology and catalysis in the broadest sense.

Reviewer #2 (Remarks to the Author):

2.1) The authors have addressed some of the issues that were raised by the three reviewers. Doing this the authors have slightly improved the manuscript. However, the major concern of both, reviewer 1 and 2, that the work only represents some optimization/slight improvements of existing technologies and reactions that have been shown extensively before has not been addressed. The yield and titers are very low. Apparently, as the authors also write in their reply, yield and titer even decrease further with increasing substrate concentration. Unfortunately no clear vision our route is given in the manuscript that would alleviate this. For example, the authors now write "Directed evolution efforts⁶³⁻⁶⁷ should undoubtedly allow improving the catalytic efficiency and turnover numbers of the individual enzymes used herein." So far directed evolution approaches gave only very modest improvements of OleT. "should undoubtedly" therefore does not appear to be valid.

Reply: see the Reply to Comment 1.2 and 1.3 above.

2.2) The point regarding the very low TTN of the Ruthenium catalyst (below 10) was not addressed.

Reply: The Ruthenium catalyst **Ru3** is a well-developed and air- and water-stable catalyst (Ref 51, *Chem. Rev.* **2010**, 110, 1746). We evaluated the metathesis of dienes **2a** and **2b** (5 mM) with **Ru3** (50 μ M, 1%) under the KP-Dod system: 75% conversion to **1a** and 88% conversion to **1b** were achieved, corresponding to 75 and 88 TTNs respectively. We surmise that the TTN of **Ru3** may be further improved by applying a higher concentration of dienes **2a** or **2b**. Unfortunately, the modest activity of UndB limits the diene concentration at the moment.

The following sentence was added to page 8:

RCM of dienes **2a** and **2b** (5 mM) in the presence of a lower catalyst **Ru3** loading (50 μ M, 1%) using the KP-Dod system afforded cycloalkenes **1a** and **1b** in 75 % and 88 % conversions respectively (corresponding to TTNs of 75 and 88).

2.3) The question of mass balance was not addressed sufficiently. Indeed there are unavoidable losses; however, these become significant simply because the total conversion is extremely low. If titers and yields were much higher these losses weren't significant anymore.

Reply: For the 3-step bis-decarboxylation-metathesis cascade (Fig. 4), the detected compounds (**1a-4a**, **1b-4b**, **1c-4c**) account for more than 90% of the initial substrate. However, as cycloalkenes are highly volatile (e.g. boiling point of **1a** is 44 °C), there are unavoidable small losses during the reactions and the workup. For reactions with oleic acid **6** or olive oil **7**, these are readily trapped in cell membranes and may be converted by native enzymes in *E. coli* (e.g. via β -oxidation). Several reaction intermediates from oleic acid **6** to diacids **4a** and **4b** (i.e., 10-hydroxystearic acid, 10-keto-stearic acid, two esters) are not commercially available and/or challenging to prepare, thus challenging their accurate quantification. Furthermore, some unidentified byproducts were observed for the reaction of **6** or **7**. Accordingly, for the extended enzyme cascades including OhyA2-MIADH-BVMO-TLL-(ChnD-ChnE), the complex reaction system precludes the unambiguous identification of the exact mass balance (**6**, **7**).

We feel however that the analytical yield of the cycloalkenes **1a-c**, determined by GC-MS using an internal standard, suffice to unambiguously highlight the potential of complex chemo-enzymatic cascades.

No change was made to the text

2.4) Taken together:

I still believe the manuscript to be impressive by the sheer amount of good work performed and that it deserves publication in a higher impact journal. I doubt it to be sufficient for Nature Communications due to the missing overarching novelty of the concept and the meager yields and titers.

Reply: Thanks for the comment. The key novelty of our proof-of-concept study is integrating organometallic catalysis with whole-cell enzyme cascades of unprecedented complexity to access a group of important chemicals (cycloalkenes, inaccessible with enzymes alone) from renewable resources. Although the yields and titers are indeed modest at this moment, a clear vision is provided on how to further enhance the efficiency and productivity of these complex cascades. Moreover, from a conceptual perspective, our study clearly illustrates the power of biocompatible metal catalysis for expanding the realm of products accessible via biocatalysis and synthetic biology. This will hopefully inspire more research in this direction and produce more diverse and valuable chemicals beyond cycloalkenes.